# Role of Polyunsaturated Fatty Acids (PUFAs) and Eicosanoids on Dry Eye Symptoms and Signs

**DOI:** 10.3390/biom14030376

**Published:** 2024-03-20

**Authors:** Simran Mangwani-Mordani, Amanda Prislovsky, Daniel Stephenson, Charles E. Chalfant, Anat Galor, Nawajes Mandal

**Affiliations:** 1Surgical Services, Department of Ophthalmology, Miami Veterans Affairs Medical Center, 1201 NW 17th Street, Miami, FL 33125, USA; smm664@med.miami.edu; 2Bascom Palmer Eye Institute, Department of Ophthalmology, University of Miami, 900 NW 17th Street, Miami, FL 33136, USA; 3Memphis VA Medical Center, 1030 Jefferson Avenue, Memphis, TN 38104, USA; 4Departments of Medicine and Cell Biology, University of Virginia School of Medicine, Charlottesville, VA 22903, USA; 5Research Service, Richmond Veterans Administration Medical Center, Richmond VA 23298, USA; 6Hamilton Eye Institute, Department of Ophthalmology, Anatomy and Neurobiology, The University of Tennessee Health Science Center, 930 Madison Ave, Memphis, TN 38163, USA

**Keywords:** eicosanoids, dry eye (DE), DE symptoms and signs, fish oil, meibomian gland dysfunction (MGD), multivitamins, polyunsaturated fatty acids (PUFAs)

## Abstract

Polyunsaturated fatty acids (PUFAs) generate pro- and anti-inflammatory eicosanoids via three different metabolic pathways. This study profiled tear PUFAs and their metabolites and examined the relationships with dry eye (DE) and meibomian gland dysfunction (MGD) symptoms and signs. A total of 40 individuals with normal eyelids and corneal anatomies were prospectively recruited. The symptoms and signs of DE and MGD were assessed, and tear samples (from the right eye) were analyzed by mass spectrometry. Mann–Whitney U tests assessed differences between medians; Spearman tests assessed correlations between continuous variables; and linear regression models assessed the impact of potential confounders. The median age was 63 years; 95% were male; 30% were White; and 85% were non-Hispanic. The symptoms of DE/MGD were not correlated with tear PUFAs and eicosanoids. DE signs (i.e., tear break-up time (TBUT) and Schirmer’s) negatively correlated with anti-inflammatory eicosanoids (11,12-dihydroxyeicosatrienoic acid (11,12 DHET) and 14,15-dihydroxyicosatrienoic acid (14,15, DHET)). Corneal staining positively correlated with the anti-inflammatory PUFA, docosahexaenoic acid (DHA). MGD signs significantly associated with the pro-inflammatory eicosanoid 15-hydroxyeicosatetranoic acid (15-HETE) and DHA. Several relationships remained significant when potential confounders were considered. DE/MGD signs relate more to tear PUFAs and eicosanoids than symptoms. Understanding the impact of PUFA-related metabolic pathways in DE/MGD may provide targets for new therapeutic interventions.

## 1. Introduction

Dry eye (DE) is a prevalent, multifactorial disease that consists of a wide range of clinical manifestations that include symptoms of ocular surface pain (characterized as “dryness”, “burning”, and “discomfort”, to name a few) and visual disturbances [1]. Signs of DE can include tear instability, insufficient tear production, and/or ocular surface disruption [2]. Closely related to DE, meibomian gland dysfunction (MGD) is defined as “a chronic, diffuse abnormality of the meibomian glands, commonly characterized by terminal duct obstruction and/or qualitative/quantitative changes in the glandular secretion”, commonly caused by epithelial gland hyperkeratinization [3]. Symptoms, such as dryness and foreign body sensations, and signs, such as tear instability, can overlap between DE and MGD [3]. Together, DE/MGD symptoms and signs impact the quality of life, negatively affect mood, and limit activities of daily living [4]. Various external and internal factors can impact DE/MGD manifestations, including weather, air pollution, diet, and systemic comorbidities [5]. Inflammation has been identified as an important intermediary between such external and internal factors and DE/MGD, and thus, many studies have examined the contributions of cellular (i.e., macrophages, regulatory T cells) [6] and soluble (i.e., tumor necrosis factor-alpha (TNF-α), interleukin-1B (IL-1B), and interleukin-6 (IL-6)) [7] mediators on disease pathophysiology. 

In this regard, the roles of pro-(omega 6, ω6) and anti-(omega 3, ω3) inflammatory polyunsaturated fatty acids (PUFAs) have been studied with respect to DE/MGD. PUFAs are eicosanoid precursors that are converted to bioactive products by three enzymatic metabolic pathways: the cyclooxygenase (COX), lipoxygenase (LOX), and cytochrome P450 (CYP) pathways [8]. The most abundant and precursor of ω6-derived eicosanoid is arachidonic acid (AA) [9], which is released from cell membranes to produce pro-inflammatory thromboxanes (i.e., TXB_2_) via the COX pathway, pro-inflammatory LOX-derived hydroxyeicosatetraenoic acids (HETEs) [8], and leukotrienes (i.e., LTB_4_) [10], which are involved in acute inflammation [11]. AA can also be metabolized into anti-inflammatory eicosanoids, such as epoxyeicosatrienoic acids (EETs), via the CYP pathway and further converted to more stable metabolites, dihydroxyeicosatrienoic acids (DHETs) [8]. ω3-derived anti-inflammatory PUFA eicosapentaenoic acid (EPA) and docosahexaenoic acid (DHA) serve as substrates for the synthesis of a different class of bioactive lipid mediators known as specialized pro-resolving mediators (SPMs) (i.e., lipoxins, resolvins, protectins, and maresins), which are involved in the resolution of inflammation [11,12].

Previously, it was suggested that chronic inflammation in DE/MGD is driven by an imbalance between ω6 (AA) and ω3 (EPA and DHA) PUFAs, leading to the hyperproduction of pro-inflammatory lipid mediators (ω6 derivatives) and the underproduction of ω3 derivatives [13,14]. In our prior study, we examined the correlations between DE/MGD features and various eicosanoids in 41 individuals. The strongest relationships were between prostaglandin E_2_ (PGE_2_) (produced via the COX pathway) and corneal staining (ρ = 0.35) and meibomian gland (MG) plugging (ρ = 0.40), *p* < 0.05 for both, indicating higher PGE_2_ levels in individuals with worse DE signs. In a similar manner, a more inflammatory PUFA profile (higher ω6: ω3 ratio) correlated with less healthy tear parameters (tear break-up time, TBUT ρ = −0.37, Schirmer score ρ = −0.38, and corneal staining ρ = 0.31) [13]. Other studies have found that both pro- and anti-inflammatory eicosanoids are increased in DE/MGD. One Singapore-based study compared individuals with poor meibum expressibility (quality ≥ 1) from ≤2 meibomian glands (case, *n* = 29) to those with normal meibum expressibilty (quality = 0) from ≥3 meibomian glands (control, *n* = 11). Cases had higher levels of pro-inflammatory eicosanoids produced by COX and/or LOX pathways compared to the controls (5-hydroxyeicosatetraenoic acid (5-HETE): 0.69 ± 0.62 vs. 0.36 ± 0.62, *p* = 0.01; leukotriene B4 (LTB_4_): 0.14 ± 0.15 vs. 0.11 ± 0.26, *p* = 0.04). However, cases also had increased levels of anti-inflammatory eicosanoids derived from EPA compared to controls (18-hydroxyeicosapentaenoic acid (18-HEPE): 0.19 ± 0.24 vs. 0.06 ± 0.04, *p* = 0.01; 12-hydroxyeicosapentaenoic acid (12-HEPE): 0.62 ± 0.66 vs. 0.52 ± 1.06, *p* = 0.03; 5-hydroxyeicosapentaenoic acid (5-HEPE): 0.12 ± 0.21 vs. 0.07 ± 0.10, *p* = 0.05) [10]. These results suggest that pro-inflammatory responses contribute to the pathophysiology of DE/MGD, with a compensatory anti-inflammatory response that aims to restore ocular surface homeostasis. 

While several studies have examined the relationships between eicosanoids produced from the LOX and COX metabolic pathways and DE/MGD, what is missing from the literature is an examination of eicosanoids produced from the AA cytochrome P450 epoxygenase pathway. Animal [15] and human studies [16] have demonstrated that cytochrome-derived eicosanoids can have potent pro- or anti-inflammatory properties, and as such, these metabolites may play a role in DE/MGD. To bridge this knowledge gap, we aimed to profile the levels of tear PUFAs, LOX, COX, and cytochrome pathway-derived eicosanoids and examine their relation to clinical symptoms and signs of DE and MGD. 

## 2. Materials and Methods

### 2.1. Study Design and Population

This prospective, single-site, cross-sectional study was conducted in accordance with the tenets of the Declaration of Helsinki, complied with the requirements of the United States Health Insurance Portability and Accountability Act (HIPAA), and was approved by the Miami Veterans Affairs (VA) Institutional Review Board. Participants were recruited from 2016 to 2017 at the Miami VA Eye Clinic, and informed consent was obtained from the subjects after an explanation of the nature and possible consequences of the study. Individuals were excluded if they had concomitant ocular or systemic conditions that could confound DE, such as anatomic abnormalities of their eyelids (i.e., ectropion), conjunctiva (i.e., pterygium), and/or cornea (i.e., Salzman’s nodular degeneration and edema); history of glaucoma, refractive, or retinal surgery; cataract surgery within the last 6 months; use of contact lenses; topical medications besides artificial tears; HIV; sarcoidosis; graft-versus host disease or a collagen vascular disease. 

### 2.2. Data Collection 

Demographic information, including age, sex, race, ethnicity, smoking, medical history, oral medications, and supplements, was collected for each patient. 

### 2.3. Ocular Symptoms

All individuals filled out validated questionnaires regarding DE symptom severity, including the five-item Dry Eye Questionnaire (DEQ-5; range: 0–22) [17] and the Ocular Surface Disease Index (OSDI; range: 0–100) [18]. Pain-specific questionnaires included a Numerical Rating Scale [19] (NRS; range: 0–10, quantifying “average intensity of eye pain during the past week”) and Neuropathic Pain Symptom Inventory-modified for the Eye [20] (NPSI-Eye) that assessed the intensity of neuropathic pain features (i.e., burning sensation, pain sensitivity to wind, light, and temperature change). 

### 2.4. Ocular Surface Assessment

All participants underwent a comprehensive ocular surface examination of both eyes, which included the following, in the order performed: (1)Measurement of tear film osmolarity (TearLAB, San Diego, CA, USA).(2)Assessment of ocular surface inflammation via InflammaDry (Quidel, San Diego, CA, USA), identifying matrix metallopeptidase 9 (MMP9) graded as 1 = present or 0 = absent based on the appearance of pink stripe.(3)Upper or lower eyelid laxity determined by rotation (0 = 0–25%, 1 = 25–50%, and 2 = 50–100%) and the snap back test (0 = prompt snapback, 1 = slowed return, and 2 = does not return fully until blinking), respectively.(4)Anterior blepharitis graded as 0 = none, 1 = mild, 2 = moderate, and 3 = severe.(5)Telangiectasias seen on the lower eyelids as 0 = none, 1 = mild vessel engorgement, 2 = moderate vessel engorgement, and 3 = severe vessel engorgement.(6)Inferior meibomian gland plugging graded as 0 = none, 1 = less than 1/3, 2 = between 1/3 and 2/3, and 3 = greater than 2/3 lid involvement.(7)Tear stability measured by placing 5 µL fluorescein in the superior conjunctivae and assessing the tear break-up time (TBUT).(8)Fluorescein corneal staining graded to the National Eye Institute (NEI) scale with five areas assessed the inferior, nasal, superior, temporal, and central, and each scored 0–3 (maximum score: 15).(9)Conjunctivochalasis in each area of the lower eyelid (nasally, medially, and temporally) graded as 0 = none, 1 = mild, 2 = moderate, and 3 = severe.(10)Tear production graded as millimeter (mm) wetting of anesthetized Schirmer’s test placed in the inferior fornix at 5 min.(11)Inferior meibomian gland drop out graded to the Meiboscale (range: 0–4) [21].(12)Meibum quality graded as 0 = clear, 1 = cloudy, 2 = granular, 3 = toothpaste, and 4 = no meibum extracted.

### 2.5. Tear Collection, PUFA, and Eicosanoid Extraction and Analysis

Schirmer strips were stored in −80 °C until analysis. Eicosanoids were extracted and analyzed by UPLC ESI-MS/MS, as previously described by us and others [22,23,24,25,26,27,28,29]. Briefly, tear strips were placed in tubes with 4 mL of water and an internal standard (IS) mixture comprising 10% methanol (400 μL) and glacial acetic acid (20 μL) and an internal standard (20 μL) containing the following deuterated eicosanoids (1.5 pmol/μL, 30 pmol total) (all standards purchased from Cayman Chemicals, Ann Arbor, MI, USA): (*d*_4_) 6keto-prostaglandin F_1_α, (*d*_4_) prostaglandin F*2*α, (*d*_4_) prostaglandin E_2_, (*d*_4_) prostaglandin D_2_, (*d*_8_) 5-hydroxyeicosatetraenoic acid (5-HETE), (*d*_8_) 12-hydroxyeicosatetraenoic acid (12-HETE), (*d*_8_) 15-hydroxyeicosatetraenoic acid (15-HETE), (*d*_6_) 20-hydroxyeicosatetraenoic acid (20-HETE), (*d*_11_) 8,9 epoxyeicosa-trienoic acid, (*d*_8_) 14,15 epoxyeicosa-trienoic acid, (*d*_8_) arachidonic acid, (*d*_5_) Eicosapentaenoic acid, (*d*_5_) docosahexaenoic acid, (*d*_4_) prostaglandin A2, (*d*_4_) leukotriene B_4_, (*d*_4_) leukotriene C4, (*d*_4_) leukotriene D4, (*d*_4_) leukotriene E4, (*d*_5_) 5(S),6(R)-lipoxin A4, (*d*_11_) 5-iPF2α-VI, (*d*_4_) 8-iso prostaglandin F2α, (*d*_11_) (±)14,15-DHET, (*d*_11_) (±)8,9-DHET, (*d*_11_) (±)11,12-DHET, (*d*_4_) prostaglandin E1, (*d*_4_) thromboxane B2, (*d*_6_) dihomo gamma linoleic acid, (*d*_5_) resolvin D2, (*d*_5_) resolvin D1 (RvD1), (*d*_5_) Maresin2, (*d*_7_) 5-OxoETE, and (*d*_5_) resolvin D3. Samples and vial rinses (5% MeOH; 2 mL) were applied to Strata-X SPE columns (Phenomenex, Torrance, CA, USA), previously washed with methanol (2 mL) and then dH_2_O (2 mL). Eicosanoids eluted with isopropanol (2 mL) were dried in vacuo and reconstituted in EtOH:dH_2_O (50:50;100 μL) prior to an ultra-high performance liquid chromatography electrospray ionization–MS/MS (UPLC ESI-MS/MS) analysis.

Eicosanoids were separated using a Shimadzu Nexera X2 LC-30AD (Shimadzu, Kyoto, Japan) coupled with a SIL-30AC auto injector (Shimadzu, Kyoto, Japan) and a DGU-20A5R (Shimadzu, Kyoto, Japan) degassing unit in the following way: A 14 min, reversed-phase LC method utilizing an Ascentis Express C18 column (150 mm × 2.1 mm, 2.7 µm) was used to separate the eicosanoids at a 0.5 mL/min flow rate at 40 °C. The column was equilibrated with 100% Solvent A (acetonitrile/water/formic acid (20:80:0.02, *v*/*v*/*v*)) for 5 min and then 10 µL of sample was injected. Further, 100% Solvent A was used for the first two min of elution. Solvent B (acetonitrile/isopropanol/formic acid (20:80:0.02, *v*/*v*/*v*)) was increased in a linear gradient to 25% Solvent B at 3 min, to 30% at 6 min, to 55% at 6.1 min, to 70% at 10 min, and to 100% at 10.10 min. Then, 100% Solvent B was held constant until 13.0 min, where it was decreased to 0% Solvent B and 100% Solvent A from 13.0 min to 13.1 min. From 13.1 min to 14.0 min, Solvent A was held constant at 100%. 

Eicosanoids were analyzed via mass spectrometric means using an AB Sciex Triple Quad 5500 Mass Spectrometer (Sciex, Toronto, Canada). Q1 and Q3 were set to detect distinctive precursor and product ion pairs. Ions were fragmented in Q2 using N2 gas for collisionally induced dissociation. The analysis used multiple-reaction monitoring in a negative-ion mode. Eicosanoids were monitored using precursor → product MRM pairs. The mass spectrometer parameters used were as follows: curtain gas: 20 psi; CAD: medium; ion spray voltage: −4500 V; temperature: 300 °C; gas 1: 40 psi; gas 2: 60 psi; declustering potential, collision energy, and cell exit potential vary per transition as reported [22,23,24,25,26,27,28,29].

### 2.6. Statistical Analysis

Statistical analyses were performed using SPSS Statistics Software version 25.0 (IBM Corp. Armonk, NY). Descriptive statistics were used to summarize participant demographics, comorbidities, medication use, DE/MGD symptoms, and signs. The normality of the distributions of variables of interest was assessed using the Shapiro–Wilk test. Given that measures were not normally distributed, Mann–Whitney U tests were run to assess the differences between medians, and Spearman correlation coefficients (ρ) were calculated to evaluate the relationship between demographics, DE signs/symptoms, pro, and anti-inflammatory markers. After inspecting residuals, linear regression models with the forward method were performed to predict the contribution of patient characteristics, comorbidities, tear PUFAs, and eicosanoids on DE/MGD symptoms and signs. *p* < 0.05 was considered statistically significant.

## 3. Results

### 3.1. Study Population

The median age of the racially diverse, predominantly male population was 63 years (interquartile range (IQR): 14) (Table 1). DE symptoms ranged from none to severe, with 90% of individuals reporting mild or greater DE symptoms as determined by a DEQ-5 ≥ 6 and 83% as determined by an OSDI ≥ 13. The majority reported some degree of ocular pain (85%, NRS ≥ 1), with 48% reporting moderate or greater pain (NRS ≥ 4). Ocular surface signs varied, with 13% displaying tear instability in the right eye (OD) (as determined by TBUT < 5 s) and 8% showing aqueous tear deficiency (as determined by Schirmer’s < 5 mm). All individuals had at least one sign of MGD, which included eyelid telangiectasias, MG plugging, or MG dropout.

### 3.2. Tear PUFAs and Eicosanoids

Several tear PUFAs and eicosanoids were collected from the study population (as seen in Appendix A) with pro-inflammatory properties (i.e., arachidonic acid (AA), thromboxane B2 (TXB_2_), 5-hydroxyeicosatetraenoic acid (5-HETE), 12-hydroxyeicosatetraenoic acid (12-HETE), 15-hydroxyeicosatetraenoic acid (15-HETE)), and anti-inflammatory properties (i.e., docosahexaenoic acid (DHA), eicosapentaenoic acid (EPA), 11,12-dihydroxyeicosatrienoic acid (11,12 DHET, 14,15-dihydroxyicosatrienoic acid (14,15, DHET)). Ratios showing the relationship between pro- and anti-inflammatory tear PUFAs were also calculated (i.e., AA:DHA, AA:EPA, ω6: ω3). Median values were calculated for all markers, including those with undetectable levels (0.001 pmol of select lipid/mg protein), and the percentage recovered indicates the frequency of detectable quantities.

### 3.3. Relationships between Tear Eicosanoids and Clinical Metrics

Mann–Whitney U tests were performed to compare differences in medians between demographics, co-morbidities, tear PUFAs, and eicosanoids. Only the significant differences are summarized in Table 2. Males had higher levels of tear pro-inflammatory eicosanoids (i.e., 12 HETE and 15-HETE) compared to females. A similar pattern was noted in Hispanic individuals (compared to non-Hispanics) and in those with diabetes (compared to non-diabetics). On the other hand, smokers had increased levels of an anti-inflammatory marker (14,15 DHET) compared to non-smokers. Also, subjects who reported taking fish oil supplements had a less inflammatory profile, with higher levels of anti-inflammatory mediators and lower inflammatory ratios compared to those not on supplements (EPA: 49.83 (IQR: 66.85) vs. 7.61 (IQR: 16.3), *p* = 0.03; AA:EPA: 35.42 (IQR: 44.18) vs. 74.11 (IQR: 75.56), *p* = 0.04). Individuals who reported taking multivitamin supplements had higher levels of both pro- (i.e., 5-HETE, AA) and anti- (i.e., 14,15 DHET, 11,12 DHET, EPA, and DHA) inflammatory eicosanoids compared to those not on supplements.

### 3.4. Relationships between Tear PUFAs, Eicosanoids, and DE/MGD Metrics

Spearman correlations were performed to examine the relationships between DE/MGD symptoms and signs and lipid mediators. Symptoms were not related to pro- or anti-inflammatory lipid mediators (Table 3). However, several significant correlations were noted with respect to tear parameters and eicosanoids. Tear stability (TBUT) negatively correlated with anti-inflammatory mediators (DHA: ρ = −0.34, *p* = 0.03 and 11,12 DHET: ρ = −0.34, *p* = 0.03). On the other hand, tear production (Schirmer) negatively correlated with both pro- (5-HETE: ρ = −0.32, *p* = 0.04) and anti (14,15-DHET: ρ = −0.40, *p* = 0.01) inflammatory eicosanoids. Finally, corneal staining positively correlated with an anti-inflammatory eicosanoid (DHA: ρ = 0.35, *p* = 0.03), all of which suggest possible compensatory mechanisms to limit inflammation-associated tear instability, tear reduction, and corneal epithelial disruption.

With regards to MGD, individuals with telangiectasias had higher pro-inflammatory lipid levels compared to those without telangiectasias (15-HETE: none: 1.07 (IQR: 1.46); mild: 1.83 (IQR: 5.53); and moderate: 2.75 (IQR: 4.05), ρ = 0.32, *p* < 0.05).

### 3.5. Linear Regression Models 

After inspecting residuals, forward linear regression models were built with significant DE/MGD signs (right eye only) from univariable analysis as dependent variables (Table 3) and all PUFAs and eicosanoids as independent variables. Other variables included in the models were demographics (i.e., gender and ethnicity), history of smoking, hypercholesterolemia, diabetes, sleep apnea, use of betablockers, anxiolytics, fish oil, and multivitamin supplements (Table 4). In all models, tear lipids remained when confounders were considered. Specifically, those with worse ocular surface parameters (lower TBUT and Schirmer, higher corneal staining, and more severe eyelid telangiectasias) had higher levels of eicosanoids with anti-inflammatory properties (11,12-DHET, 14,15-DHET, and DHA) (Table 4, Figure 1).

## 4. Discussion

In this study, we found some relationships between DE/MGD signs, PUFAs, and their derivatives, even when considering potential confounders. Specifically, the multivariable analysis found negative correlations between tear film parameters (stability and production) and anti-inflammatory eicosanoids (11,12-DHET and 14,15-DHET), suggesting compensatory mechanisms to a pro-inflammatory state driven by tear abnormalities. Similar findings were noted when considering corneal staining and eyelid telangiectasias, as they were related to higher levels of the anti-inflammatory eicosanoid DHA. 

Our results share similarities and differences from prior studies. The similarities include detecting COX-pathway-derived pro-inflammatory eicosanoids (i.e., 5-HETE) in tears. A case (*n* = 40)–control (*n* = 30) study in subjects with a mean Schirmer value of 15.5 ± 14.0 mm, non-invasive TBUT (NIBUT) of 7.34 s, and MGD (defined as eyelid telangiectasias ≥ 1 (range: 0–3), MG plugging ≥ 1 (range: 0–3), or meibum expressibility ≥ 1 (range: 0–3)) showed that low tear production related to a pro-inflammatory state, given the negative correlations between Schirmer values and pro-inflammatory eicosanoids (5-HETE: ρ = −0.51, *p* = 0.017, and 9-HETE: ρ = −0.78, *p* < 0.001) [10]. Our univariable results, noting a negative correlation between Schirmer and 5-HETE, coincide with prior findings. Yet, studies have also detected increased anti-inflammatory metabolites derived from the CYP metabolic pathway of DHA [30], including 20-HDoHE (coefficient = −0.02, *p* value < 0.001) and 17-HDoHE (coefficient = −0.00, *p* value = 0.001), in subjects with lower tear production. The relationship between pro- (i.e., 5-HETE) and anti- (i.e., 20-HDoHE and 17-HDoHE) inflammatory markers may again represent a compensatory response of eicosanoids to counteract the pro-inflammatory state that occurs in aqueous deficiency. Our study findings were similar in that tear stability (TBUT) and tear production (Schirmer) were negatively related to the anti-inflammatory eicosanoids 11,12-DHET and 14,15-DHET, respectively. 

The findings with respect to corneal staining also share similarities and differences with the literature. In some studies, corneal staining was positively related to various pro-inflammatory prostaglandins (i.e., PGE_2_ (ρ = 0.35, *p* < 0.05) [13] and PGF_2α_ (coefficient = 0.09, *p* value = 0.02)) [10]. At the same time, corneal staining has also been positively correlated with various anti-inflammatory eicosanoids (18-HEPE (coefficient = 0.07, *p* value = 0.02), 20-HDoHE (coefficient = 0.13, *p* value = 0.03), and 17-HDoHE (coefficient = 0.03, *p* value = 0.01)) [10]. In a similar manner, we found significant relationships between more severe corneal staining and increased levels of DHA. While differences in study populations, DE/MGD definitions, and extraction methods may have contributed to the variable findings across studies, some eicosanoid signatures emerge in relation to DE/MGD signs, including prostaglandins, DHA, and HETEs.

A unique finding in our study was the detection of tear eicosanoids metabolized via the CYP pathway (i.e., 11,12 DHET, 14,15-DHET). Studies outside the eye have related DHET molecules to pro-inflammatory conditions such as coronary heart disease (CHD), with individuals with CHD having higher levels of 14,15-DHET compared to controls (2.53 ± 1.60 ng/L vs. 1.65 ± 1.54 ng/L, *p* = 0.036) [16]. We similarly found significant correlations between low tear stability and production and DHET molecules, indicating that higher levels of anti-inflammatory eicosanoids may be involved in disease modulation and restoration of homeostasis across disease states and organs [16]. 

Beyond tear and ocular surface parameters, several patient-related factors, including gender, ethnicity, the presence of comorbidities, and smoking, are related to levels of various tear eicosanoids, with ethnicity and diabetes remaining in multivariable models. Specifically, males, Hispanics, and individuals with diabetes had higher levels of pro-inflammatory eicosanoids (i.e., 12-HETE and 15-HETE) compared to their counterparts, while current or previous smokers had higher levels of an anti-inflammatory eicosanoid (i.e., 14,15-DHET) compared to non-smokers. These findings share similarities and differences with prior literature. While previous studies have not found differences in tear lipids by gender [10,13], mixed findings were noted by ethnicity, with higher [31], lower [31], and similar [32,33] levels of plasma anti-inflammatory PUFAs reported. Similar to our study, increased pro-inflammatory and angiogenic eicosanoids (i.e., LTB4 and 12-HETE) were noted in individuals with chronic diseases such as diabetes [34] and CHD [35]. Interestingly, prior studies have also noted increased plasma pro- (i.e., 5-HETE) and anti- (i.e., 11,12-DHET and 14,15-DHET) inflammatory eicosanoids in smokers, suggesting that compensatory mechanisms may occur both in blood and tears [36]. Although the aim of our study was not to describe the impact of oral supplements such as fish oil and multivitamins on tear composition, we noted that individuals who reported taking supplements had a less inflammatory tear film (lower AA: EPA, AA: DHA and ω6: ω3) compared to their counterparts. Similar findings were noted in our prior study with respect to individuals taking ω3 (DHA and EPA) supplements [13]. While the results of studies on the benefits of ω3 and ω6 supplementation in DE have been mixed [37,38,39,40,41,42], our findings suggest the need for further research on this topic.

As with all studies, our findings need to be considered in light of the limitations of this study, which included its cross-sectional nature, predominately male population, limited sample size, and sensitivity of the tear eicosanoid assay. Furthermore, the origin of lipid mediators in tears is not known, with possible sources, including the MG, lacrimal gland, and/or ocular surface epithelium. Finally, our assay did not include some pro-resolving eicosanoids, such as resolvins, maresins, and neuroprotectin D1. Despite these limitations, in this study, we found both pro- and anti-inflammatory markers in subjects with tear abnormalities and MGD, which is similar to prior studies. We also detected new findings, such as CYP-derived metabolites, which have been reported to possess anti-inflammatory properties. 

## 5. Conclusions

Overall, our findings highlight the complexity of studying tear eicosanoids in DE/MGD, as their presence may contribute to or be a compensatory mechanism for an abnormal ocular surface environment. It is plausible that the ocular surface constantly autoregulates itself to re-establish homeostasis, and thus, longitudinal studies are needed to evaluate our findings more robustly. A better understanding of the role of eicosanoids in DE/MGD is needed, as this knowledge may improve treatment algorithms by suggesting which medications (i.e., corticosteroids that block phospholipase A_2_ enzyme preventing production of AA and all downstream products vs. nonsteroidal anti-inflammatory drugs (NSAIDs) that only block the COX pathway) would be optimal in an individual patient [43]. Moreover, beyond COX and LOX inhibition, selective cytochrome P-450 inhibitors may have a beneficial role, which needs to be further defined in the DE/MGD field.

## Figures and Tables

**Figure 1 biomolecules-14-00376-f001:**
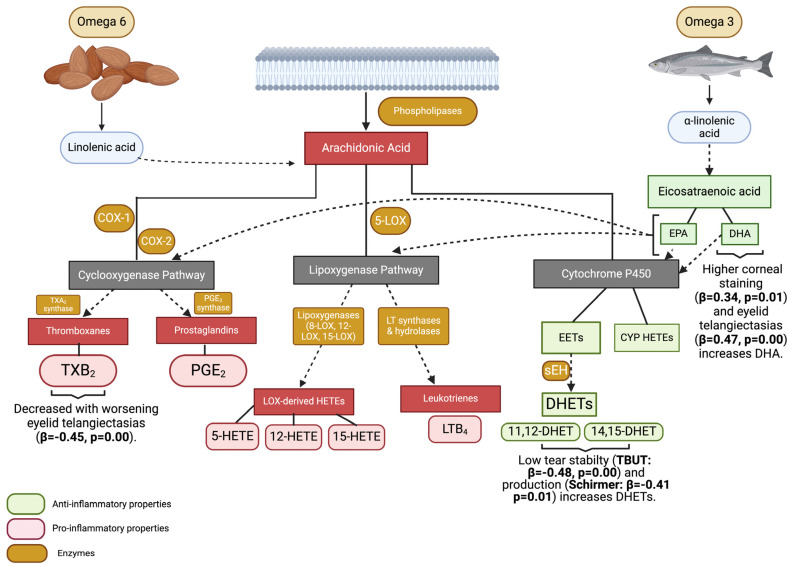
Graphic representation of relationships between PUFAs, eicosanoids, and dry eye signs. COX-1: cyclooxygenase 1; COX-2: cyclooxygenase 2; TXA_2_: thromboxane A_2_; PGE_2_: prostagladin E_2_; TXB_2_: thromboxane B_2_; 5-LOX: 5-lipoxygenase; 8-LOX: 8-lipoxygenase; 12-LOX: 12-lipoxygenase; 15-LOX: 15-lipoxygenase; LT: leukotrienes; LOX: lipoxygenase; 5-HETE: 5-hydroxyeicosatetraenoic acid; 12-HETE: 12-hydroxyeicosatetraenoic acid; 15-HETE: 15-hydroxyeicosatetraenoic acid; LTB_4_: leukotriene B_4_; EPA: eicosapentaenoic acid; DHA: docosahexaenoic acid; EETs: epoxyeicosatrienoic acids; she: soluble epoxide hydrolases; DHETs: dihydroxyeicosatrienoic acids; 11,12 DHET: 11,12-dihydroxyeicosatrienoic acid; 14,15 DHET: 14,15-dihydroxyicosatrienoic acid; CYP HETEs: cytochrome P450 hydroxyicosatetraenoic acids. Overall, ocular surface parameters were more abnormal in individuals with higher anti-inflammatory properties, suggesting compensatory responses to an adverse ocular surface environment. Figure created with BioRender.com.

**Table 1 biomolecules-14-00376-t001:** Demographics and clinical information of the study population.

Characteristics	Frequencies
**Demographics, % (*n*)**	
Sex, male	95% (38)
Race, White	30% (12)
Ethnicity, non-Hispanic	85% (34)
**Comorbidities, % (*n*)**	
Smoking, current	35% (14)
Hypertension	68% (27)
Hypercholesterolemia	63% (25)
Diabetes	40% (16)
PTSD	30% (12)
Depression	65% (26)
Osteoarthritis	55% (22)
Sleep apnea	35% (14)
BPH	18% (7)
**Medications, % (*n*)**	
Betablockers	18% (7)
Statins	53% (21)
Antidepressants	65% (26)
Anxiolytics	63% (25)
Antihistamines	20% (8)
NSAIDs	33% (13)
ASA	43% (17)
Fish oil supplements	10% (4)
**Devices, % (*n*)**	
CPAP	15% (6)
**Dry eye symptoms and ocular pain, median (IQR)**	
DEQ-5	12.5 (7.0)
OSDI	34.4 (37.6)
NRS of average pain 1 week (0–10)	3.0 (4.0)
NPSI-Eye total (0–100)	21.5 (29.8)
*** Dry eye signs, % (*n*)**	
Tear osmolarity, mOsm/L, *median (IQR)*	299.5 (15.8)
MMP-9, =1, (0–1)	60% (24)
Upper lid laxity, ≥2, (0–2)	25% (10)
Lower lid laxity, ≥2, (0–2)	18% (7)
Anterior blepharitis, ≥2, (0–3)	5% (2)
Telangiectasias, ≥2 (0–3)	10% (4)
Inferior meibomian gland plugging, ≥2, (0–3)	45% (18)
TBUT, seconds, *median* (IQR),	9.5 (6.7)
Corneal staining, ≥2 (0–15)	45% (18)
Conjunctivochalasis, ≥2, (0–3)	58% (23)
Schirmer’s, mm, *median* (IQR)	11.5 (13)
Meibomian gland dropout, ≥2, (0–4)	58% (23)
Meibum quality, ≥2, (0–4)	50% (20)

PTSD: post-traumatic stress disorder; BPH: benign prostatic hyperplasia; NSAIDs: nonsteroidal anti-inflammatory drugs; ASA: acetyl-salicylic acid; CPAP: continuous positive airway pressure; DEQ-5: 5-Item Dry Eye Questionnaire; OSDI: Ocular Surface Disease Index; NRS: Numerical Rating Scale; NPSI-Eye: Neuropathic Pain Symptom Inventory Modified for the Eye; MMP-9: matrix metallopeptidase 9; TBUT: tear break-up time; IQR: interquartile range. * Values taken from the right eye.

**Table 2 biomolecules-14-00376-t002:** Significant differences in medians between demographics, medical history, and eicosanoids.

Group	Eicosanoid	Median (IQR), (pmol Select Lipid/mg Protein]) *	*n*	Mann–Whitney U	*p*-Value
**Demographics**					
Gender					
Males	12-HETE	3.02 (3.10)	38	6.00	0.05
Females		0.72 (0)	2		
Ethnicity					
Non-Hispanics	15-HETE	1.06 (1.44)	34	43.00	0.03
Hispanics		2.80 (3.49)	6		
**Comorbidities**					
Smoking					
Yes **	14,15 DHET	0.10 (0.12)	33	50.00	0.02
No		0.05 (0.08)	7		
Hypercholesterolemia					
Yes	TXB_2_	0.00 (26.66)	25	119.00	0.05
No		20.73 (52.11)	15		
Diabetes					
Yes	12 HETE	4.10 (3.47)	16	117.00	0.04
No		2.18 (2.51)	24		
Sleep apnea					
Yes	AA: DHA	6.24 (2.49)	14	112.00	0.05
No		7.09 (3.94)	26		
Yes	ω6: ω3	5.67 (2.44)	15	107.00	0.03
No		6.81 (8.95)	26		
**Medications**					
Betablockers					
Yes	15 HETE	4.65 (4.60)	7	46.00	0.01
No		1.07 (1.39)	33		
Anxiolytics					
Yes	TXB_2_	16.89 (46.6)	25	112.00	0.03
No		0.00 (16.60)	15		
Fish oil supplements					
Yes	5 HETE	2.00 (1.16)	4	28.00	0.05
No		0.76 (0.72)	36		
Yes	EPA	49.83 (66.85)	4	23.00	0.03
No		7.61 (16.3)	36		
Yes	AA: EPA	35.43 (44.18)	4	26.00	0.04
No		74.11 (75.56)	36		
Yes	AA: DHA	4.96 (2.33)	4	25.00	0.03
No		7.15 (3.01)	36		
Yes	ω6: ω3	4.30 (2.80)	4	18.00	0.02
No		6.81 (2.31)	36		
Multivitamin supplements					
Yes	14,15 DHET	0.10 (0.16)	21	123.50	0.04
No		0.08 (0.11)	19		
Yes	11,12 DHET	0.05 (0.07)	21	103.00	0.00
No		0.00 (0.03)	19		
Yes	5 HETE	1.21 (1.51)	21	84.00	0.00
No		0.70 (0.79)	19		
Yes	EPA	15.47 (28.44)	21	109.50	0.02
No		4.01 (10.85)	19		
Yes	DHA	138.60 (197.55)	21	100.00	0.00
No		54.22 (68.68)	19		
Yes	AA	742.38 (1377.13)	21	119.00	0.03
No		412.64 (735.95)	19		
Yes	AA: EPA	53.96 (34.25)	21	125.50	0.05
No		102.87 (80.17)	19		

IQR: interquartile range; 12-HETE:12-hydroxyeicosatetraenoic acid; 15-HETE:15-hydroxyeicosatetraenoic acid; 14,15 DHET:14,15-dihydroxyeicosatrienoic acid; TXB_2_: thromboxane B2; 5 HETE: 5-hydroxyeicosatetraenoic acid; AA: arachidonic acid; DHA: docosahexaenoic acid; ω6: pmega-6 fatty acid; ω3: pmega-3 fatty acid; EPA: eicosapentaenoic acid; 11,12 DHET:11,12dihydroxyeicosatrienoic acid. 

 pro-inflammatory; 

 anti-inflammatory; 

 pro/anti-inflammatory ratio. * Unit of each lipid is normalized with the amount of protein in the sample as a pico mole of lipid per milligram of protein (pmol/mg protein). ** Includes history of smoking and current smokers.

**Table 3 biomolecules-14-00376-t003:** Spearman correlations demonstrating relationships between DE/MGD symptoms and signs and pro- and anti-inflammatory markers.

	Pro-Inflammatory					Anti-Inflammatory				Pro/Anti-Inflammatory Ratios		
	AA	TXB_2_	5 HETE	12 HETE	15 HETE	DHA	EPA	11,12 DHET	14,15 DHET	AA: DHA	AA: EPA	ω6: ω3
	*ρ*	*ρ*	*ρ*	*ρ*	*ρ*	*ρ*	*ρ*	*ρ*	*ρ*	*ρ*	*ρ*	*ρ*
**Symptoms**												
DEQ-5	−0.11	0.11	0.02	0.12	0.07	−0.12	−0.01	0.10	0.01	0.14	0.20	0.15
OSDI	−0.04	0.10	−0.00	−0.03	0.04	−0.11	−0.02	0.01	0.04	0.24	0.14	0.22
Avg. eye pain intensity over 1 week	0.01	0.14	0.08	0.13	0.01	−0.05	0.08	0.13	0.18	0.13	−0.05	0.11
NPSI-Eye total	−0.05	0.12	0.06	0.08	0.00	−0.11	0.04	0.05	0.12	0.16	−0.04	0.14
**§ Signs**												
Tear osmolarity	0.18	−0.02	0.14	−0.01	0.02	0.30	0.24	0.10	−0.01	−0.25	−0.34	−0.31
Upper lid laxity	−0.02	−0.27	0.14	0.24	0.08	0.02	0.01	0.12	−0.01	−0.17	0.13	−0.14
Lower lid laxity	−0.05	−0.25	0.02	0.13	0.02	0.01	0.06	−0.05	−0.06	−0.20	−0.09	−0.22
Anterior blepharitis	0.14	−0.01	0.03	0.08	0.24	0.11	0.03	0.11	−0.04	0.09	0.29	0.09
Eyelid telangiectasias	0.24	−0.18	0.17	−0.06	**0.32 ***	0.25	0.21	−0.07	0.06	−0.09	−0.00	−0.09
Meibomian gland plugging	0.11	0.16	0.03	−0.01	−0.12	0.12	0.07	0.03	0.08	0.09	−0.03	0.05
TBUT	−0.29	−0.16	−0.13	0.17	−0.05	**−0.34 ***	−0.29	**−0.34 ***	−0.14	0.05	0.12	0.05
Staining	0.30	0.10	0.25	0.13	0.17	**0.35 ***	0.23	0.26	0.26	−0.05	−0.05	−0.06
Conjunctivochalasis	0.01	0.05	0.01	0.19	−0.14	0.02	0.10	0.18	0.18	0.03	−0.17	−0.01
Schirmer	−0.30	−0.14	**−0.32 ***	−0.20	−0.24	−0.31	−0.26	−0.22	**−0.40 ****	−0.09	0.01	−0.07
Meibomian glands drop out	0.17	−0.29	0.19	−0.01	0.08	0.15	0.17	−0.01	0.15	0.09	0.11	0.09
Meibum quality	0.19	−0.01	0.13	0.02	0.06	0.22	0.28	−0.18	0.08	−0.07	−0.04	−0.12

AA: arachidonic acid; TXB_2_: thromboxane B2; 5-HETE: 5-hydroxyeicosatetraenoic acid; 12-HETE: 12-hydroxyeicosatetraenoic acid; DHA: docosahexaenoic acid; EPA: eicosapentaenoic acid; 11,12 DHET: 11,12-dihydroxyeicosatrienoic acid; 14,15 DHET: 14,15-dihydroxyicosatrienoic acid; ω6: omega 6; ω3: omega 3; DEQ-5: 5-Item Dry Eye Questionnaire; OSDI: Ocular Surface Disease Index; NPSI-Eye: Neuropathic Pain Symptom Inventory modified for the eye; TBUT: tear break-up time. ** *p* < 0.01, * *p* < 0.05. § Values taken from the right eye only.

**Table 4 biomolecules-14-00376-t004:** Linear regression models examining impact of tear PUFAs, eicosanoids, demographics, and comorbidities on DE/MGD symptoms and signs.

		Unstandardized Coefficients		Standardized Coefficients	Sig.	95% Confidence Intervals		Coefficient of Determination
	Models	B	SE	β	*p* Value	Lower	Upper	Adjusted R^2^
**DE Signs**								
Eyelid telangiectasias	DHA	0.00	0.00	0.47	0.00	0.00	0.01	0.36
	TXB_2_	−0.01	0.00	−0.45	0.00	−0.02	−0.00	
	Ethnicity	0.54	0.25	0.29	0.04	0.03	1.05	
TBUT	11,12-DHET	−65.0	19.6	−0.48	0.00	−104.64	−25.30	0.54
	AA: DHA	0.99	0.35	0.41	0.00	0.29	1.70	
Staining	DM	1.93	0.63	0.43	0.00	0.66	3.19	0.30
	DHA	0.01	0.00	0.34	0.02	0.00	0.01	
Schirmer	14,15-DHET	−29.66	10.76	−0.41	0.01	−51.45	−7.87	0.17

DE: dry eye; AA: arachidonic acid; DHA: docosahexaenoic acid; TXB_2_: thromboxane B2; TBUT: tear break-up time; 11,12 DHET:11,12dihydroxyeicosatrienoic acid; 14,15 DHET:14,15-dihydroxyeicosatrienoic acid; EPA: eicosapentaenoic acid. 

 pro-inflammatory; 

 anti-inflammatory; 

 pro/anti-inflammatory ratio.

## Data Availability

The data presented in this study are available on request from the corresponding author.

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
