# Peer review of "Role of Polyunsaturated Fatty Acids (PUFAs) and Eicosanoids on Dry Eye Symptoms and Signs"

_biomolecules, 2024, doi:10.3390/biom14030376_

Round 1

Reviewer 1 Report

Comments and Suggestions for Authors

The manuscript titled “Role of Polyunsaturated Fatty Acids (PUFAs) and Eicosanoids on Dry Eye Symptoms and Signs” by Simran Mangwani-Mordani et al. studies the profile of tear PUFA metabolites and their relationships with symptoms and signs of dry eye and meibomian gland dysfunction in a clinical trial. The work has a correct introduction to the field and well presented and analyzed results. However, there are minor concerns that must be addressed:

Line 158: The meaning of the abbreviation “IS” is not indicated.

Table 2: The differences obtained in 12-HETE between men and women should be discussed.

Table 2: Why are the levels of an anti-inflammatory such as 14,15 DHET higher in smokers than in non-smokers?

Figure 1: The abbreviations indicated in the legend should have the same format as in the tables.

Reviewer 2 Report

Comments and Suggestions for Authors

Summary

In this study by Mangwani-Mordani et al., the relationships between tear polyunsaturated fatty acids (PUFAs) and eicosanoids with dry eye (DE) and Meibomian gland dysfunction (MGD) symptoms and signs were investigated. The authors found no correlation between tear PUFAs and eicosanoids with DE and MGD symptoms. On the other hand, there were significant correlations between DE and MGD signs with the anti-inflammatory eicosanoids 11,12-DHET and 14,15-DHET, the anti-inflammatory PUFA DHA and the pro-inflammatory eicosanoid 15-HETE. The authors postulate that the higher levels of anti-inflammatory eicosanoids may be due to compensatory mechanisms to limit inflammation due to tear abnormalities.

Overall comments

The study setup is straightforward and sound and the analysis supports the authors’ conclusions. The discussion section is well-written.

Specific comments

1. Supplemental Table 1, Table 2: The units for the median values of tear lipids and eicosanoids should be indicated in the column headings or table legends.

2. Line 264, “Lower tear stability (TBUT) was negatively correlated with…”: This sentence can be confusing to readers. I think the authors mean that a lower TBUT is correlated with higher levels of anti-inflammatory mediators?

3. Lines 290 – 294, 304 – 305: Since the independent variables are the levels of PUFAs and eicosanoids and the dependent variables are DE and MGD signs, the regression models are predicting ocular surface parameters based on PUFA and eicosanoid levels. But the phrasing of the text seems to suggest the reverse, that the authors are predicting PUFA and eicosanoid levels based on ocular surface parameters.
